# Prevalence of urogenital and intestinal schistosomiasis among school children in South-west Nigeria

Johnson A. Ojo[1], Samuel A. Adedokun[1], Akeem A. Akindele[1], Adedolapo B. Olorunfemi[1], Olawumi A. Otutu[1], Taiwo A. Ojurongbe[2], Bolaji N. Thomas📝[3], Thirumalaisamy P. Velavan[4,5‡], Olusola Ojurongbe📝[1‡*]

1 Department of Medical Microbiology & Parasitology, Ladoke Akintola University of Technology, Ogbomoso, Nigeria, 2 Department of Mathematical Sciences, Osun State University, Osogbo, Nigeria, 3 Department of Biomedical Sciences, College of Health Sciences and Technology, Rochester Institute of Technology, Rochester New York, United States of America, 4 Institute of Tropical Medicine, University of Tübingen, Tübingen, Germany, 5 Vietnamese German Center for Medical Research (VG-CARE), Hanoi, Vietnam

‡ These authors are joint senior authors on this work.
* oojurongbe@lautech.edu.ng

**Data Availability Statement:** The data is available at https://data.mendeley.com/datasets/ddn8k3b5jj/1 doi: 10.17632/ddn8k3b5jj.1.

## Abstract

### Background

The risk of co-infection with *Schistosoma haematobium* and *S. mansoni* and the potential harmful effect on morbidity and control is enhanced by the overlapping distribution of both species in sub-Saharan Africa. Despite the reported high endemicity of both species in Nigeria, studies on the spread and effect of their mixed infection are limited. Therefore, a cross-sectional survey was conducted among school children in two communities in South-west Nigeria to investigate the prevalence of mixed human schistosome infection, intensity, and possible ectopic egg elimination.

### Methods

Urine and stool samples were collected from consenting school children in Ilie and Ore communities of Osun State, Nigeria. *Schistosoma haematobium* eggs were detected in urine using the urine filtration technique, while *S. mansoni* eggs were detected in stool using the Kato–Katz thick smear technique.

### Results

The study enrolled 466 primary and secondary school children (211; 45.3% males vs. 255; 54.7% females; mean age 11.6 ± 3.16 years). The overall prevalence of schistosomiasis was 40% (185/466), with 19% (89/466) recording single *S. haematobium* infection while 9% (41/465) had a single *S. mansoni* infection. The geometric mean egg count for *S. haematobium* was 189.4 egg/10ml urine; 95% CI: range 115.9–262.9, while for *S. mansoni*, it was 115.7 epg; 95% CI: range 78.4–152.9. The prevalence of ectopic *S mansoni* (*S. mansoni* eggs in urine) was 4.7%, while no ectopic *S. haematobium* (*S. haematobium* eggs in stool) was recorded. Mixed infection of *S. haematobium*/*S. mansoni* had a prevalence of 9.5%

**Funding:** The author(s) received no specific funding for this work.

**Competing interests:** The authors have declared that no competing interests exist.

(44/466). More females (54.5%) presented with *S. haematobium*/*S. mansoni* co-infection. For both parasites, males had higher infection intensity, with a significant difference observed with *S. haematobium* (p = 0.0004). Hematuria was significant in individuals with single *S. haematobium* infection (p = 0.002), mixed ectopic *S. haematobium*/*S. mansoni* (p = 0.009) and mixed *S. haematobium*/*S. mansoni*/ectopic *S. mansoni* (p = 0.0003).

## Conclusions

These findings suggest the probability of interspecific interactions between *S. haematobium* and *S. mansoni*. Scaling up of mass administration of praziquantel and control measures in the study areas is highly desirable.

## Author summary

In sub-Saharan Africa, human schistosomiasis is a neglected disease of public health concern caused mostly by *Schistosoma haematobium* and *Schistosoma mansoni*. The overlapping range of both species in Africa considerably increases the chance of co-infection. School-aged children are the most vulnerable, as they participate in water contact activities that expose them to free-swimming cercariae released by infected snail species in freshwater. This study examined the probable mixed human *Schistosoma* infections and associated disease variables in school children in the communities of Ilie and Ore in southwest Nigeria. This study reveals a high prevalence of mixed *S. haematobium* and *S. mansoni*, and ectopic *S. mansoni* eggs (*S. mansoni* eggs in urine) elimination, highlighting the possible ongoing control challenges in this area. Furthermore, this study indicates that some form of inter-specific interaction exists between *S. haematobium* and *S. mansoni*, and may produce potentially significant consequences for developing morbidity in the study areas.

## Introduction

Schistosomiasis, a neglected tropical disease targeted for elimination by the World Health Organization (WHO) [1], is a significant public health problem, with Nigeria [2–4] ranking first among African countries with the highest disease burden [5]. The disease caused by the genus *Schistosoma* is responsible for the most obvious reduction in age-standardized years lived with disability (YLD) between 2006 and 2016 [6]. The most affected group are school-aged children, involved in water contact activities that brings them in contact with the free-swimming cercariae, released from infected snail species in freshwater [3,7]. The disease is present in 78 countries, affecting more than 250 million people annually, presenting with two major forms; a urogenital disease caused by *S. haematobium* and intestinal disease caused by *S. mansoni*, *S. japonicum*, *S. mekongi*, and *S. intercalatum*. *S. haematobium* (*Sh*) and *S. mansoni* (*Sm*) are the two major species endemic in sub-Saharan Africa, along with a few cases of *S. intercalatum*, localized in rain forest areas of Central Africa [1,8].

Urogenital schistosomiasis is associated with outcomes, such as hematuria, bladder cancer, and hydronephrosis, while the chronic intestinal disease is characterized by hepatomegaly, splenomegaly, and progressive periportal fibrosis resulting in portal hypertension, esophageal varices, liver surface irregularities, portal-systemic venous shunts, and hematemesis [9,10].

The impaired physical and cognitive development arising from chronic infection among children is a major concern in many parts of the world [11]. The risk of co-infection with *Sh* and *Sm* is greatly enhanced by the overlapping distribution of both species in Africa [12]. However, there is a lack of information on the determinants, distribution and the impact of such mixed infections on endemic populations. Results from co-infection in experimental models show the two species could form heterologous male-female pairs, with the male carrying the female to its preferred site for oviposition and the female producing eggs characteristic of her species in an uncharacteristic site [13,14]. This phenomenon, referred to as hybridization, is believed to be responsible for ectopic egg elimination resulting in the detection of *Sm* eggs in urine or *Sh* eggs in feces, in areas where mixed infection occurs [15,16], and suggestive of possible sexual interaction in nature between *Sh* and *Sm*. Hybridization of human schistosomes species was recently observed in France from a patient who had just returned from Côte d'Ivoire [17,18]. It is believed that disease epidemiology and phenotypic characteristics could be altered by hybridization, which could ultimately affect the parasite's transmission and host compatibility [19,20].

An increasing number of foci where co-infections between *Sh* and *Sm* occur has been reported in some parts of Africa [15,21–23]. In some of these foci, differences in schistosomiasis-associated morbidity, as well as infection intensity, have been reported between single and mixed infections [23]. The presence of one species may alter the course of infection or disease caused by the other, and such interactions could arise through competition for nutrients or mates or immune-mediated mechanisms, including cross-reactive immune responses. Also, an increase in *S. mansoni* infection has been observed after praziquantel treatment in co-endemic areas [23,24]. These interspecific interactions of *Sh/Sm*, although not fully understood, may have important implications on epidemiology, associated morbidity, and control measures, including PZQ treatment. Information on the occurrence and prevalence of mixed infections will help us answer critical questions on the underlying mechanisms toward morbidity and develop effective strategies for preventing and controlling schistosomiasis in co-endemic areas. Both *Sh* and *Sm* occur in Nigeria, with *Sh* having a higher prevalence. While several studies have reported both species in the same foci, there has been very little information on the prevalence and impact of mixed *Sh* and *Sm* infection and the possible ectopic egg elimination.

In developing countries, children aged 5–17 years are at the highest risk of infection and are the most infected group targeted with preventive mass chemotherapy. Also, in the context of the current WHO strategies for controlling and eliminating schistosomiasis, our study focused on school-aged children. Therefore, this study presents a cross-sectional report among school children to investigate possible mixed *Schistosoma* infections and associated disease covariates in two schistosomiasis endemic communities in Nigeria.

## Material and methods

### Ethics statement

The Ethical Review Committee, Osun State Ministry of Health approved the study (approval number OSHREC/PRS/569T/131). Verbal and/or written informed consent was obtained from the parents/guardians and assent from the participants before they were recruited.

### Study site

The study was conducted in Ore and llie communities, Osun State, Nigeria. The two communities are located very closely on latitude 4˚34' and 4˚36'E, and Longitude 7˚56' and 7˚58'N, and only separated by a dam in the rain forest zone. The dam, owned and managed by the

State Water Corporation, Olorunda local government area, southwest Nigeria, is the breeding site of *Schistosoma* due to the abundant presence of the snail intermediate host in the dam. These communities depend on the dam for their domestic water supply, fishing, and other water-related activities. There have been previous reports of schistosome endemicity in these communities [25].

### Study population, inclusion, and exclusion criteria

The study population consisted of primary and secondary school children aged 4–19 years attending Ore Community Primary and Secondary School and Ilie Community Primary and Secondary school. They are in Primary 2–5, Junior Secondary School 1–3, and Senior Secondary School 1–3. Participants with a history of being clinically ill during recruitment and those who used antischistosomal drugs in the last six months before the study and those whose parents refused to give informed consent and children who refused to give assent were excluded.

### Study design and sample size calculation

The study design was cross-sectional and conducted between March 2018 and May 2019. Headteachers and principals of the selected primary and secondary schools were duly informed by the school board about the study. The significance of the study was highlighted to the teachers, parents and the school children before sample collection. The sample size was obtained using the formula for a cross-sectional study [26]. Using a prior prevalence of 37.5% among school children positive for schistosomiasis [25],a marginal error of 5%, and a type 1 error of 5%, a minimum sample of 289 school children was needed.

### Sample collection

In all, 466 school children participated in this study. Individual demographic information was collected with a structured questionnaire, while two sterile, universal containers, individually labeled for urine and stool collection, were distributed to consenting school children. Instructions on the procedure for collecting urine and stool were given to the students to ensure that contamination was avoided. For each participant, one urine and stool sample were collected.

### Parasitological examination

The presence of *Sh* eggs was detected using the urine filtration technique, as previously described [3]. Briefly, 10 ml of the freshly passed urine sample was pushed through a micro-filter membrane of 10–12 μm (MF, Whatman, New Jersey, USA) using a syringe. The micro-filter membrane was then carefully placed on a glass slide, mounted on a microscope, and examined using a light microscope's low-power objective (10×). For stool analysis, two Kato–Katz thick smears were prepared using 41.7 mg templates of the stool material for each and microscopically examined for *Sm* and other intestinal parasites [27]. Slides were examined by two independent and experienced scientists. For quality control, 15% of all positive and negative slides were re-examined by a third independent microscopist who was blind to the first two scientist results. The *Sh* infection intensity was expressed as the number of eggs detected in 10ml of urine (eggs/10ml), while *Sm* infection intensity was expressed as the number of eggs detected per gram of feces (epg). The counted eggs were categorized into a light infection (1–99 epg for *Sm* and 1–49 eggs per 10ml of urine for *Sh*), moderate (100–399 epg for *Sm*), and heavy infections ($\geq$400epg for *Sm* and $\geq$ 50 eggs per 10ml of urine for *Sh*) [28]. Single infection was defined as passing eggs of only one species, and mixed infection as passing eggs of

both *Sm* and *Sh*. The incidence of ectopic egg excretion was measured qualitatively (positive/negative). Ectopic egg elimination refers to detecting schistosomal eggs via the unusual route–i.e., *Sh* eggs in feces or *Sm* eggs in urine. Overall *Sm* infection refers to both mixed and single *Sm* infections. Overall *Sh* infection includes both mixed and single *Sh* infections. Each child found to be positive for any of the schistosome species was treated with 40mg/kg praziquantel by the study team.

## Data analysis

Data were double entered into an excel sheet, cleaned, and then analyzed using IBM Statistical Package for Social Sciences (SPSS) for Windows version 20 (SPSS, Inc., Armonk, USA). Data were described using percentages, geometric means, and 95% confidence interval. The egg output data was 10 log-transformed to normalize skewed egg distribution. Geometric means of egg count (GM epg or eggs per 10 ml of urine) were computed for microscopically positive individuals, and intensity of infection was analyzed. The $\chi$-square test was used to evaluate the association between infection status (*Sm*, *Sh*, and mixed infection) and disease covariates (sex, age, etc.). The independent-samples t-test was used to compare GM infection intensities with age and sex.

## Results

A sample comprising 5 grams of feces and 10 ml of urine was obtained from 466 primary and secondary school children who participated in the study. Complete parasitological data were also obtained from all these participants. The participants consisted of 211 (45.3%) males and 255 (54.7%) females with a mean age of 11.6±3.16 years. The overall mean weight and height are 31.2±9.60kg and 1.41±0.78m, respectively. According to age group, the breakdown of the infection showed that the older age group (12–19 years) was generally more infected except for children with mixed ectopic *Sm*/ *Sh* infection. In all the age group comparisons, no significant difference was observed. Similarly, no significant difference was observed between gender and infection prevalence, with females having a higher proportion of *Sh* (57.3%), *Sm* (58.5%) single infections, and *Sh*/*Sm* mixed infection (54.5%) while males had higher prevalence for the mixed infections of ectopic *Sm*/*Sh* (80.0%) and ectopic *Sm*/*Sh*/*Sm* (54.5%) (Table 1). 33.7% of the participants positive for *Sh* had blood in their urine, while 66.3% were *Sh* positive but without blood in their urine, and the difference was statistically significant (*p = 0.002)*. Similarly, the proportion of the mixed infections of ectopic *Sm*/*Sh* (*p = 0.009)* and ectopic *Sm*/*Sh*/*Sm* (*p = 0.0003)* had a statistically significant effect on the proportion of participants with blood in their urine. The mean weight and the mean height of the study population are shown in Table 1. Mean weight and mean heights of positive and negative participants showed no statistically significant difference. The overall prevalence of schistosomiasis in the study was 40% (185/466). Single *Sh* infection among the participants was 19% (89/466) with a geometric egg count of 189.4egg/10mls (95%CI: 115.9–262.9), while 9% (41/465) had single *Sm* infection with a geometric egg count of 115.7 epg (95%CI: 78.4–152.9). Mixed *Sh*/*Sm* infection was recorded in 9.5% (44/466) of the study population. Mixed ectopic *Sm* occurring along with *Sh* (Fig 1) was recorded in 4.5% (21/466) of the study population while 1(0.2%) participant had a single ectopic *Sm* infection. The occurrence and distribution of *Schistosoma* infection are shown in Table 2. An overall prevalence of 31% and 10% for *Sh* and *Sm*, respectively, was observed.

The association between ectopic *Sm* egg elimination and infection intensities of *Sh* and *Sm* is shown in Table 3. A high prevalence of ectopic *Sm* egg was observed in high infection intensities of both *Sh* (18%) and *Sm* (15.6%), producing a significant association in both cases. Fig 2

**Table 1. General characteristics and prevalence of human schistosomiasis in the study population.**

| Characteristics | Total Population | Sh (single infection) n = 89 | Sm (single infection) n = 41 | Sh/Sm (Mixed infection) n = 33 | ESm (single infection) n = 1 | ESm/Sh (Mixed infection) n = 10 | Sh/Sm/ ESm (Mixed infection) n = 11 |
|---|---|---|---|---|---|---|---|
| **Mean age ± SD** | 11.6±3.16 | 12.0±3.10 | 12.5±2.70 | 12.3±2.50 | - | 11.3±2.21 | 12.8±3.5 |
| **Mean weight ± SD** | 31.2±9.60 | 32.9±9.96 | 32.0±9.10 | 33.8±7.74 | | 29.1±5.07 | 33.4±10.40 |
| **Mean Height± SD** **Age group** | 1.41±0.78 | 1.41±0.16 | 1.69±1.98 | 1.40±0.15 | | 1.37±0.07 | 1.39±0.16 |
| **4–11 years (%)** | 221 (47.4) | 40 (45.0) | 15 (36.6) | 11 (33.3) | 0 | 6 (60.0) | 4 (36.4) |
| **12–19 years (%)** | 245 (52.6) | 49 (55.1) | 26 (63.4) | 22 (66.7) | 1 (100) | 4 (40.0) | 7 (63.6) |
| **p-value** | | 0.638 | 0.189 | 0.105 | - | 0.528 | 0.550 |
| **Sex** | | | | | | | |
| **Male (%)** | 211 (45.3) | 38 (42.7) | 17 (41.5) | 15 (45.5) | 0 | 8 (80.0) | 6 (54.5) |
| **Female (%)** | 255 (54.7) | 51 (57.3) | 24 (58.5) | 18 (54.5) | 1 (100) | 2 (20.0) | 5 (45.5) |
| **p-value** | | 0.636 | 0.627 | 1.000 | - | 0.050 | 0.556 |
| **Blood in Urine** | | | | | | | |
| **Present** | 98 (21.0) | 30 (33.7) | 6 (14.6) | 11 (33.3) | 0 | 6 (60.0) | 8 (72.7) |
| **Absent** | 368 (79.0) | 59 (66.3) | 35 (85.4) | 22 (66.7) | | 4 (40.0) | 3 (27.3) |
| **p-value** | | 0.002* | 0.421 | 0.079 | | 0.009* | 0.0003* |

**Key:** *Sh*: *Schistosoma haematobium*; *Sm*: *Schistosoma mansoni*; E*Sm*: Ectopic *Schistosoma mansoni*
*Significant p<0.05

shows the relationship between age prevalence and infection intensity in the study population. Age group 12–19 years recorded a higher prevalence of *Sh*, *Sm*, and ectopic *Sm* infection than the younger age group (4–11 years), but the difference in all cases was not statistically

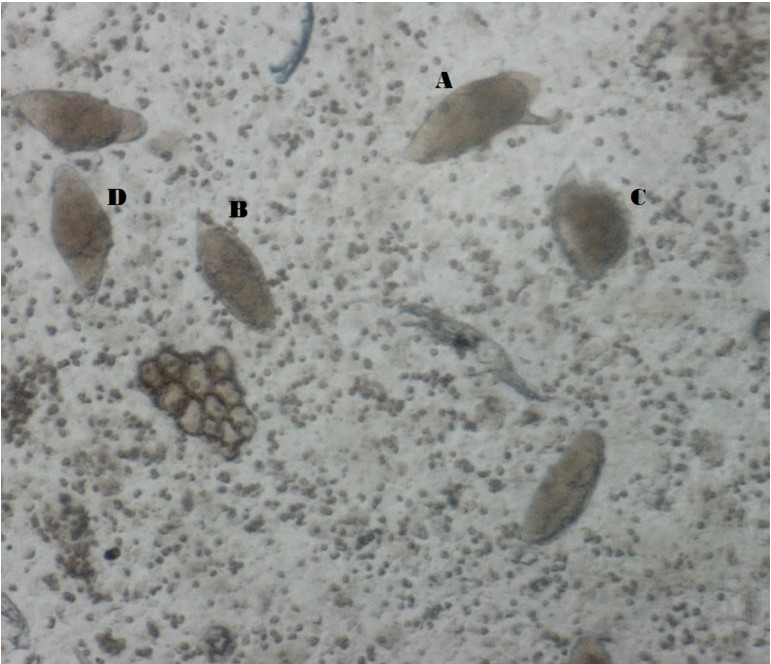

**Fig 1.** Ectopic egg elimination of *S. mansoni* in the urine of one of the study participants: **A**: *S. mansoni* egg**; B, C, D**: *S. haematobium*.

**Table 2. Schistosomal infection prevalence and intensities.**

| | *Sm* infection | | *Sm* infection | | Prevalence n = 466 (%) | *Sh* infection intensity | | *Sm* infection intensity | |
|---|---|---|---|---|---|---|---|---|---|
| | Urine | Stool | Urine | stool | | GM egg/ 10 ml | (95% CI) | GM epg | (95% CI) |
| Positive participants | | | | | **185 (40.0)** | | | | |
| **Single infection** | + | - | - | - | 89 (19.1) | 189.4 | 115.9–262.9 | | |
| | - | + | - | - | 0 | | | | |
| | - | - | + | - | 1 (0.2) | | | | |
| | - | - | - | + | 41 (9.0) | | | 115.7 | 78.4–152.9 |
| **Mixed Infections** | | | | | **44 (9.5)** | **668.6** | **395.4–941.8** | **229.2** | **100.5–357.9** |
| E*Sm* Infection | | | + | | **21 (4.7)** | | | | |
| Negative participants | | | | | **281 (60.4)** | | | | |
| Overall *Sh* infections | | | | | 143 (30.8) | 399.4 | 263.7–535.2 | | |
| Overall *Sm* infections | | | | | 85 (18.3) | | | 174.4 | 105.6–243.3 |

**Key**: *Sh*: *Schistosoma haematobium*; *Sm*: *Schistosoma mansoni*; E*Sm*: Ectopic *Schistosoma mansoni*

+ = Positive;— = Negative

significant. In both *Sh* and *Sm*, the younger age group (4–11 years) had higher infection intensity and was statistically significant ($p$ = 0.016) in the *Sm* group. The pattern was different for the ectopic *Sm* infection group as the older age group recorded the higher infection intensity, but the difference was not statistically significant (Fig 2).

The relationship between sex and infection intensity in the study population is shown in Fig 3. Females were more infected with both *Sh* and *Sm*, but the difference was not statistically significant. On the other hand, males recorded more ectopic *Sm* infection, but the difference was not statistically significant. For both *Sh* and *Sm*, males had higher infection intensity, and the difference was significant in those infected with *Sh* (p = 0.0004). In ectopic *Sm*, females had higher infection intensity, but the difference was not statistically significant (Fig 3).

## Discussion

We present the analysis of mixed *Sh* and *Sm* infections and the ectopic egg elimination of schistosome eggs in school children in Osun State, Nigeria. The study revealed a high

**Table 3. Relation between ectopic *Schistosoma mansoni* eggs in urine and intensities of S. haematobium egg in urine and S.mansoni egg in stool.**

| Intensities of *Sh* in urine (eggs/10ml) | *Sm* eggs in urine | | |
|---|---|---|---|
| | N | Cases | Prevalence (%) |
| 0 | 323 | 1 | 0.3 |
| 1–9 | 0 | 0 | 0 |
| 10–49 | 32 | 1 | 3.1 |
| $\geq$ 50 | 111 | 20 | 18 |
| Total | 466 | 22 | |
| **Intensity of *Sm* in stool (epg)** | | | |
| 0 | 381 | 11 | 2.9 |
| 1–99 | 53 | 6 | 11.3 |
| $\geq$ 100 | 32 | 5 | 15.6 |
| Total | 466 | 22 | |

**Key**: *Sh*: *Schistosoma haematobium*; *Sm*: *Schistosoma mansoni*

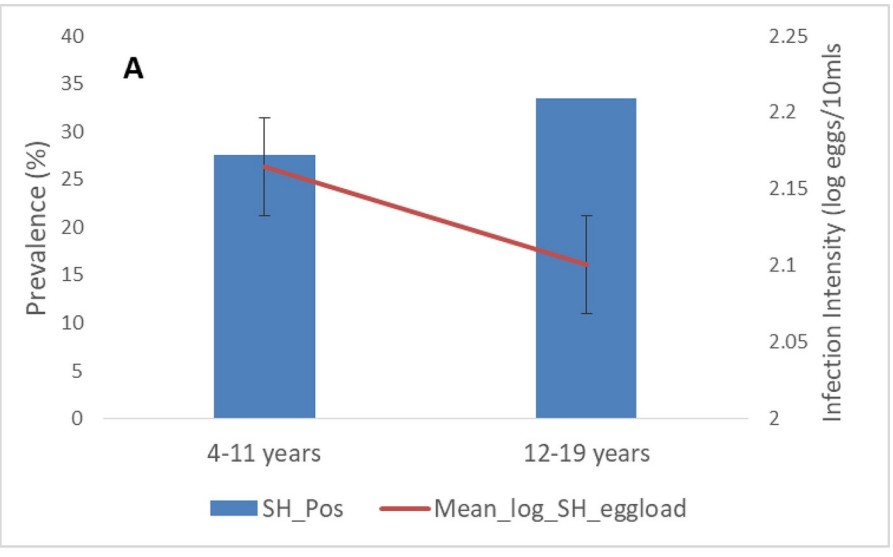

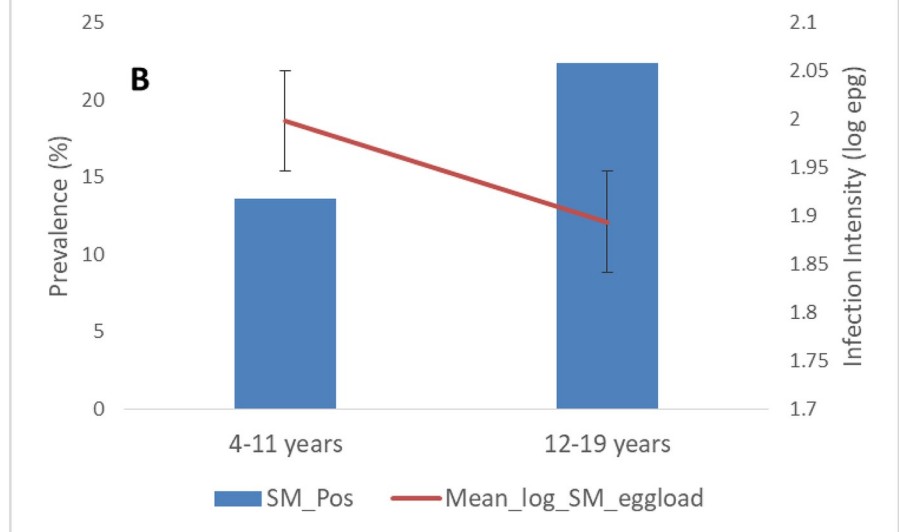

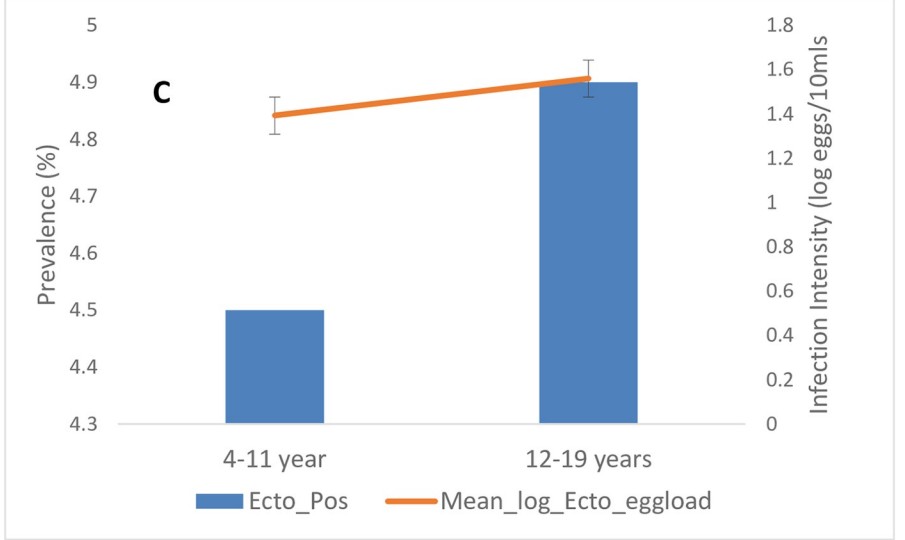

**Fig 2. Age-prevalence and intensity curves for schistosomiasis.** The bars indicate overall infection prevalence per age group. Lines indicate mean log-transformed infection intensities among positive subjects. **A**: *S. haematobium* infection; Age vs prevalence $p > 0.05$; infection intensity vs age $p = 0.55$. **B:** *S. mansoni* infection; Age vs prevalence **p = 0.016**\*; infection intensity vs age p = 0.33. **C:** Ectopic egg *(S. mansoni* in urine) infection; Age vs prevalence $p > 0.05$; infection intensity vs age $p = 0.059$.

prevalence of both *Sh* and *Sm* infections among school children in Ilie and Ore communities of Osun State, Nigeria. The overall prevalence of schistosomiasis was 40%, and as expected, the prevalence of *Sh* (31%) was significantly higher than *Sm* (10%) ($p < 0.05$). An earlier report had shown widespread urinary schistosomiasis in the Niger River basin, the Southwest, the Central and Northern Highlands, and Lake Chad. At the same time, intestinal schistosomiasis was less prevalent and widespread in Nigeria [29]. Across the different regions of Nigeria, mixed *Sh* and *Sm* infection prevalence ranges from 60.8 to 4.8% and 8.9 to 2.9%, respectively [30–33]. The high prevalence of both *Sh* and *Sm* reported in this study highlights the possible ongoing control challenges in this area. Also worrisome is the observation that 10% of the school children were co-infected with both *Sh* and *Sm*. Inter-specific parasite interactions in areas with mixed species infections have been predicted to impact host morbidity significantly. For example, lower liver morbidity has been reported in individuals with mixed infection than those with single *Sm* infections, and higher bladder morbidity reported in those with mixed compared to those with single *Sh* infections [23]. The lowering impact of liver morbidity in individuals with mixed infections was suggested to be caused by the hybrid eggs produced by the mating of *Sh* males with *Sm* females. For instance, the deposition of such eggs in the urinary oviposition site (ectopic egg elimination) reduces the amount of classical *Sm* eggs capable of inducing liver morbidity [23,34]. While this study observed *Sm* eggs in urine, *Sh* eggs were not recovered in the stool. By implication, the high prevalence and high intensity of *Sh* co-infection with *Sm* in this study may aggravate the associated *Sh* bladder morbidity. To clarify this observation, a future study must investigate the impact of mixed *Sh* and *Sm* co-infections on both liver and bladder morbidity and other schistosome-related clinical manifestations in the study area.

The occurrence of *S. mansoni* lateral spined egg (4.7%) ectopic excretion was observed in the urine of study participants. Ectopic egg elimination (*Sh* eggs in feces and *Sm* eggs in urine) has been reported in endemic areas where both schistosomes co-exist [15,35]. The presence of ectopic infection in sympatry warrants intensification of monitoring (diagnosis) and control to avert the emergence of or curtail the spread of a possible hybrid of *S. mansoni* and *S. haematobium*. Hybridization resulting from closely related sister species of schistosomes like *S. bovis* (that causes intestinal schistosomiasis in ruminants) and *Sh*, has been reported in Senegal [36,37] and is also linked to the outbreak of schistosomiasis in Corsica, France [19]. Similarly, hybridization between the two major human schistosomes, *Sh* and *Sm*, which used to be very rare or not taken into consideration, possibly because of the assumption of the significant phylogenetic distance, has now been described in Senegal [38] and in France in a patient that originated from Côte d'Ivoire [18]. Although our study did not conduct a hybridization study, the occurrence of ectopic *Sm* could imply hybridization between the two human schistosomes as previously reported in Cameroon [15,16] and may warrant further investigation. The emergence of hybrids may negatively impact schistosomiasis control as they are well adapted to intermediate hosts, modify the epidemiology of the disease [17,39,40], spread to new areas, and become invasive populations [19].

Significantly, we observed more individuals within the age groups 4-11years and 12–19 years with lower infection intensities among those concurrently infected with both species compared to single infections, contrary to reports elsewhere [34,35]. The heterospecific

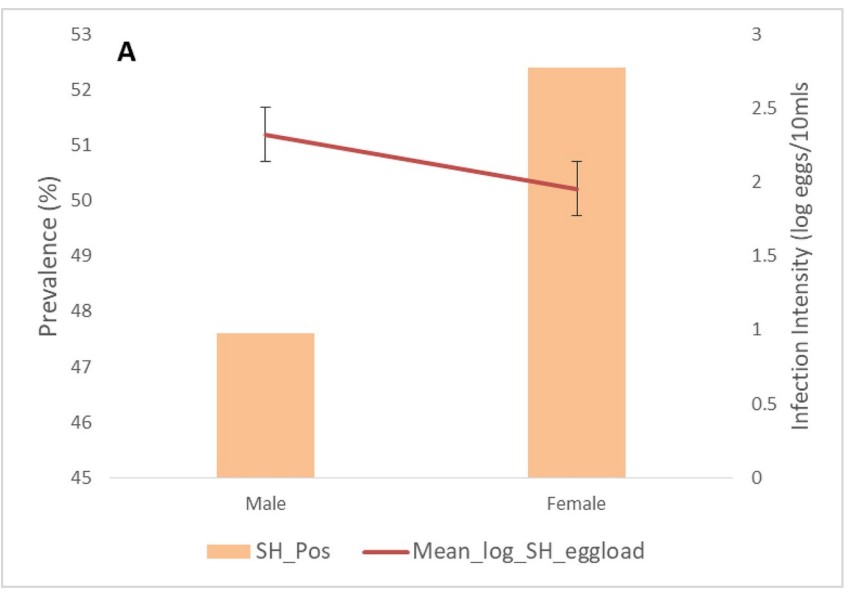

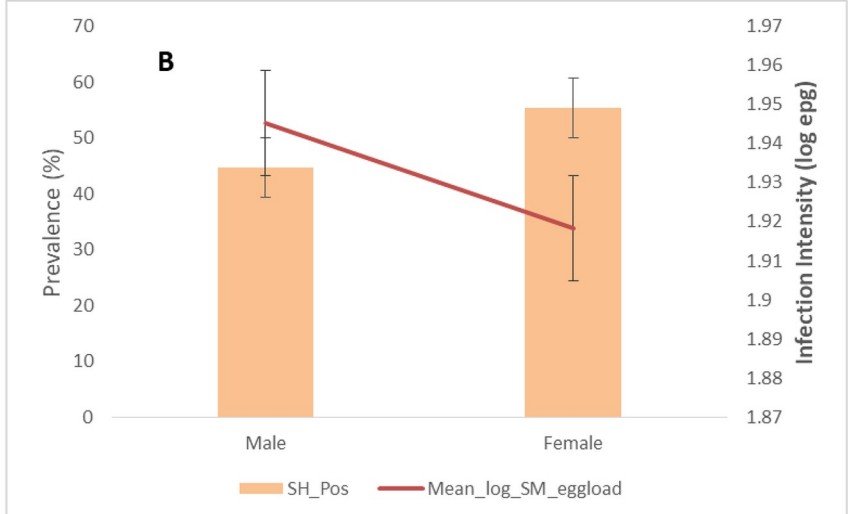

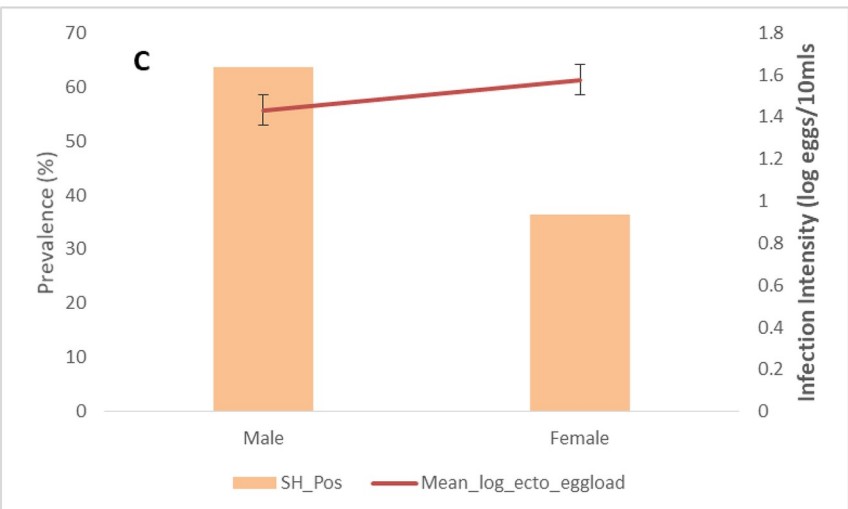

**Fig 3. Sex-prevalence and intensity curves for schistosomiasis.** The bars indicate overall infection prevalence per sex. Lines indicate mean log-transformed infection intensities among positive subjects. **A**: *S. haematobium* infection; Sex vs prevalence $p = 0.487$; infection intensity vs sex $\boldsymbol{p = 0.004}$*. **B**: *S. mansoni* infection; Age vs prevalence $p = 0.883$; infection intensity vs age $p = 0.797$. **C:** Ectopic egg *(S. mansoni* in urine) infection; Age vs prevalence $p = 0.073$; infection intensity vs age $p = 0.289$.

interaction between *Sm* and *Sh* and the high egg output in the 4–11 years age range could have severe or moderate consequences on the associated morbidity outcomes [15,23,34]. Potentially, heterospecific pairs could determine the egg type produced, its pathogenicity, and affect disease transmission. Therefore, additional information is needed about egg production and egg viability for concurrent infections. Further research in other endemic communities where there is an overlap of the two species is recommended to better understand the biomedical and public health consequences of mixed infection.

The age-related prevalence of schistosomiasis has been shown to increase as the age increases, peaking in adolescence and lowering among adults [41]. Unfortunately, adults were not included in this study, making it impossible to investigate this age-infection profile. Nonetheless, the adolescent group was significantly more infected but had a lower intensity of infection than the younger age group in this study. The older children are engaged in more water contact activity leading to the observed higher prevalence, but possess a long history of exposure and higher parasite-specific acquired immunity leading to lower infection intensity [41]. The prevalence of *Sh* and *Sm* was higher in females, and the males, on the other hand, had a higher prevalence of mixed infections. Both *Sh* and *Sm* recorded higher infection intensity in males, while for mixed infection, the infection intensity was higher in females. Earlier studies have documented heavier infection in males than females in *Sh* and *Sm*, contrary to our observation [42,43], although others have agreed [44,45]. Socio-cultural or behavioral factors focusing mainly on differences in the water contact pattern between males and females are generally implicated in the frequently observed gender-related differences in prevalence and infection intensity [3], although susceptibility factors like hormonal differences and genetic factors cannot be ruled out. The explanation for the differences in gender-related prevalence in this study may not be precise, but we speculate that females had higher water contact activities with a considerably longer duration of body exposure. The higher infection intensity observed in males may warrant further investigation as it is generally believed that high testosterone levels in males will significantly lower the infection prevalence and intensity [46]. Since this is not the focus of our study, it is clear that more studies will be needed to decipher the impact of gender on infection prevalence and intensity in our study area.

A close association was observed between hematuria and *Sh* eggs in the urine, similar to the reports of Ekpo et al. (2010) [4]. The close relationship between hematuria and the presence of eggs in the urine could be explored to assess urinary schistosomiasis in communities (after eliminating other conditions that could precipitate blood in urine) as it may also be helpful in determining endemicity in areas where urine microscopy might not be possible.

Understanding the exact relationship between mixed infection and infection intensity is crucial, as increased egg loads can significantly affect morbidity [12]. Higher *Sh* and *Sm* infection intensities were recorded in mixed than in single infections, and a positive association between *Sh* and *Sm* infections was reported in this study. While some studies have reported higher infection intensities in mixed infections [35,47], other studies on a larger scale have reported inconsistent results [48,49]. Possibly, the relationship between mixed infection and infection intensity varies according to local differences in *Sm* and *Sh* transmission. Therefore, larger sample size in different locations might be needed to accurately decipher the influence of mixed schistosome infection on the infection intensity.

The collection of only one stool and urine sample from each study participant in this study may have underestimated the infection prevalence and intensity. It is recommended that two or more consecutive samples should be collected for increased accuracy. Unfortunately, due to logistics and cost, multiple samples could not be collected in this study. However, we posit that the current observations provide enough evidence to encourage further studies and influence decision-making. We recommend further research in other endemic communities where there is an overlap of the two species to better understand the biomedical and public health consequences by including some clinical examinations (e.g., palpation and ultrasonography) not considered in this study.

In summary, this study reveal a high prevalence of mixed *Sh* and *Sm;* and ectopic *Sm* eggs elimination in Ilie and Ore communities of Osun State Nigeria. Furthermore, this study indicates that some form of inter-specific interaction exists between *Sh* and *Sm*, and may produce potentially significant consequences for developing morbidity in the study areas. Therefore, further study on the impact of mixed *Sh* and *Sm* infections on both liver and bladder morbidities and scaling up of mass administration of praziquantel and control efforts in the study areas are highly recommended.

## Acknowledgments

The authors are sincerely grateful to all consenting participants, their parents, and all the teachers for their cooperation. OO appreciate the continuous support of Alexander von Humboldt Foundation, Germany. The image of the egg of *Schistosoma* was captured with the assistance of Dr. Adrian Streit in his laboratory at Max Planck Institute for Evolutionary Biology, Tuebingen

## Author Contributions

**Conceptualization:** Olusola Ojurongbe.

**Data curation:** Taiwo A. Ojurongbe.

**Formal analysis:** Johnson A. Ojo, Samuel A. Adedokun, Akeem A. Akindele, Adedolapo B. Olorunfemi, Olawumi A. Otutu.

**Investigation:** Johnson A. Ojo, Samuel A. Adedokun, Akeem A. Akindele.

**Methodology:** Samuel A. Adedokun, Akeem A. Akindele, Adedolapo B. Olorunfemi, Olawumi A. Otutu.

**Supervision:** Bolaji N. Thomas, Thirumalaisamy P. Velavan, Olusola Ojurongbe.

**Writing – original draft:** Johnson A. Ojo.

**Writing – review & editing:** Bolaji N. Thomas, Thirumalaisamy P. Velavan, Olusola Ojurongbe.

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
