## [Decision Letter · Decision Letter 0]

23 Mar 2021

Dear Dr Ojurongbe,

Thank you very much for submitting your manuscript "Epidemiology of mixed urogenital and intestinal schistosomiasis among school children in two endemic communities of Southern Nigeria" for consideration at PLOS Neglected Tropical Diseases. As with all papers reviewed by the journal, your manuscript was reviewed by members of the editorial board and by several independent reviewers. In light of the reviews (below this email), we would like to invite the resubmission of a significantly-revised version that takes into account the reviewers' comments. 

We cannot make any decision about publication until we have seen the revised manuscript and your response to the reviewers' comments. Your revised manuscript is also likely to be sent to reviewers for further evaluation.

Sincerely,

Mike J Doenhoff

Associate Editor

Christine Budke

Deputy Editor

Reviewer's Responses to Questions

**Key Review Criteria Required for Acceptance?**

**Methods**

-Are the objectives of the study clearly articulated with a clear testable hypothesis stated?

-Is the study design appropriate to address the stated objectives?

-Is the population clearly described and appropriate for the hypothesis being tested?

-Is the sample size sufficient to ensure adequate power to address the hypothesis being tested?

-Were correct statistical analysis used to support conclusions?

-Are there concerns about ethical or regulatory requirements being met?

Reviewer #1: The methods used are ok, but my worry is on the sample collection.

Reviewer #2: The objectives of the study are not clear.

In the methods, the authors do not explain why the study was conducted only in school children. In the results, two groups are presented according to age, but in the methods this division is not presented.

The collection of only one urine sample and one stool sample for each student may have limited the finding of positive samples in the study: why is this point not discussed by the authors?

The samples were collected from two communities, but the results are presented as a single sample. What is the reason for this choice? Was the prevalence similar in both communities? For both species of Schistosoma?

In the statistical analysis, was a normality test applied to the sample to choose the appropriate comparison test between the variables.

It is known that the intensity of schistosoma infection is affected by different factors, and not just age and sex. Variables such as hygiene habits, number and time of exposure to water collections, sanitation, previous infection and previous parasitic treatment, could also have been analyzed in the epidemiological study. Why were they not analyzed?

The absence of morbidity analysis in co-infection is a limiting factor in the study.

The authors gave no information about the study period.

Reviewer #3: The objectives of the study are not clearly articulated

The study design is not described

Though the sample size is big enough, the population is only partially described

the sample size is big enough

The statistical analysis is not appropriately reported

There are no ethical issues

**Results**

-Does the analysis presented match the analysis plan?

-Are the results clearly and completely presented?

-Are the figures (Tables, Images) of sufficient quality for clarity?

Reviewer #1: The analysis are okay and the results presented well. The issue is on the signs on table 2 which were not explained

Reviewer #2: The figures shown are not in high resolution. The variables that present significant results could be presented in the figure itself and not in the legend.

Table 3 is confused for the interpretation of the results.

It is necessary to explain the set of abbreviations: 'ectopic Sm/Sh' and 'ectopic Sm/Sh/Sm'

Reviewer #3: No specific comments

**Conclusions**

-Are the conclusions supported by the data presented?

-Are the limitations of analysis clearly described?

-Do the authors discuss how these data can be helpful to advance our understanding of the topic under study?

-Is public health relevance addressed?

Reviewer #1: Authors should remove sections on hybridization as pointed out in the text

All my other comments are inserted as notes in the text

Reviewer #2: When introducing the manuscript, the authors point out that hybridization between schistosoma species has not yet been reported in Africa (line 79). However, in lines 238 and 239 of the discussion, they present 2 studies that report the ectopic presence of schistosoma eggs in Cameroon (reference 14) and Senegal (reference 33). It may therefor be appropriate to rewrite the introduction paragraph.

The discussion of the article should be reviewed. It is superficial and limited. 

At the beginning of the article, the authors suggest that the encounter of ectopic eggs is unprecedented in the study area. But throughout the article, there is decreasing emphasis on this observation and the discussion does not present a differential result. If it is really an unprecedented result, I recommend modifying the article to highlight this finding.

As this is an epidemiological study, it would be interesting to add a paragraph on the importance of the findings for local public health, according to the guidelines for coping with schistosomiasis as an important public health problem by the WHO.

Reviewer #3: The limitations are not clearly described

**Editorial and Data Presentation Modifications?**

Reviewer #1: Minor revision

Reviewer #2: (No Response)

Reviewer #3: (No Response)

**Summary and General Comments**

Reviewer #1: The study is good and deserves publication, however the authors should revise the use of words "impact" since the study is basically about prevalence

Reviewer #2: (No Response)

Reviewer #3: The manuscript entitled "Epidemiology of mixed urogenital and intestinal schistosomiasis among school children in two endemic communities of Southern Nigeria"submitted by Ojo et al. is important, and could give insight into the Schistosomiasis control challenges in areas where both major species are endemic.

However, the manuscript has a substantial limitations which the authors have to address prior any further consideration.

While the title clearly indicates an epidemiological study, there is no clear study design and nor any sampling frame which may permit a validation of the findings. The sites description indicated a previous study reporting the presence of S. hematobium, in urine and not the S. mansoni species. The statistical test used to test association, is not reported appropriately. All together the manuscript requires a deep revision.

In details:

Abstract section:

It is not clear what the authors called "ectopic egg": for non experts in schistosomiasis a brief explanation is required here.

The study design and methodology are missing, as well as brief study procedures, as appropriate in the Abstract.

Prevalence and intensity are confusedly reported and should be separated.

Would hematuria have been associated with S. hematobium?

Introduction Section:

• There is poor background information regarding clinical or epidemiological consequences of co-infection of S.h /S.m as well as ectopic eggs. Please provide more information to justify the research question.

MM section:

• Study design is missing

• Sample size calculation and sample collection should be separated

• Study procedure as well as inclusion and exclusion criteria are missing

• While the sample size calculated was 289, the authors recruited 466 participants: is there an explanation? 

• Is there any reason for use of sterile container to collect stool and urine?

Results section:

• What does '5gm' mean?

• Will mixed infection Sh/Sm differ from Sm/Sh?

• Can author provide the odds CI:95% of hematuria in S.h positive versus S.h negative?

Discussion Section

1. Authors should cross check the p-value which should be <0.05 instead of >0.05

2. Authors should provide more specific information on lower liver morbidity and higher bladder morbidity

3. What do the authors mean by this: "Consequently, the collection of urine specimens and their examination may not be necessary in the classification of communities according to the level of endemicity of urinary schistosomiasis".?

PLOS authors have the option to publish the peer review history of their article (what does this mean?). If published, this will include your full peer review and any attached files.

Reviewer #1: Yes: Prof Sammy Olufemi Sam-Wobo

Reviewer #2: No

Reviewer #3: No
---

## [Editor Report · Decision Letter 1]

11 Jun 2021

Dear Dr Ojurongbe,

Thank you very much for submitting your manuscript "Prevalence of urogenital and intestinal schistosomiasis among school children in South-west Nigeria" for consideration at PLOS Neglected Tropical Diseases. As with all papers reviewed by the journal, your manuscript was reviewed by members of the editorial board and by several independent reviewers. The reviewers appreciated the attention to an important topic. Based on the reviews, we are likely to accept this manuscript for publication, providing that you modify the manuscript according to the review recommendations. 

The authors have made extensive amendments to the manuscript, following referees' suggestions.

I have re-read the manuscript and incorporated/suggested a few minor alterations, most of which can probably be attended to in-house.

Sincerely,

Mike J Doenhoff

Associate Editor

Christine Budke

Deputy Editor

The authors have made extensive amendments to the manuscript, following referees' suggestions.

I have re-read the manuscript and incorporated/suggested a few minor alterations, most of which can probably be attended to in-house.

Figure Files:

Data Requirements:

Reproducibility:

References

---

## [Editor Report · Decision Letter 2]

5 Jul 2021

Dear Dr Ojurongbe,

We are pleased to inform you that your manuscript 'Prevalence of urogenital and intestinal schistosomiasis among school children in South-west Nigeria' has been provisionally accepted for publication in PLOS Neglected Tropical Diseases.

Best regards,

Mike J Doenhoff

Associate Editor

Christine Budke

Deputy Editor

---

## [Editor Report · Acceptance letter]

19 Jul 2021

Dear Dr Ojurongbe,

We are delighted to inform you that your manuscript, "Prevalence of urogenital and intestinal schistosomiasis among school children in South-west Nigeria," has been formally accepted for publication in PLOS Neglected Tropical Diseases.

Best regards,

Shaden Kamhawi

co-Editor-in-Chief

Paul Brindley

co-Editor-in-Chief
